# Evaluation of the performance of nucleic acid amplification tests (NAATs) in detection of chlamydia and gonorrhoea infection in vaginal specimens relative to patient infection status: a systematic review

Minttu M Rönn,[1,2] Louise Mc Grath-Lone,[1] Bethan Davies,[1] Janet D Wilson,[3] Helen Ward[1]

Poster presented at 3rd Joint Conference of the British HIV Association, BHIVA with the British Association for Sexual Health and HIV, BASHH. Liverpool United Kingdom. Abstract: HIV Medicine 15 (pp 109-110), 2014.

For numbered affiliations see end of article.

**Correspondence to**
Dr Minttu M Rönn;
mronn@hsph.harvard.edu

## ABSTRACT

**Objective** We evaluated the performance of nucleic acid amplification tests (NAATs) using vaginal specimens in comparison to specimens from the cervix or urine in their ability to detect chlamydia and gonorrhoea infection in women based on patient infection status (PIS).

**Design** Systematic review.

**Data sources** EMBASE and Ovid MEDLINE databases were searched through 3 October 2017.

**Eligibility criteria for selecting studies** We included studies that tested samples from the vagina and ≥1 other site (cervix and/or urine) with ≥2 NAATs for chlamydia and ≥2 NAATs or 1 NAAT and culture for gonorrhoea for each site.

**Data extraction and synthesis** Performance is defined as the sensitivity of a NAAT using a specimen site and PIS of the patient. We assessed risk of bias using modified QUADAS-2.

**Results** Nine publications met the inclusion criteria (eight for chlamydia; six for gonorrhoea) and were narratively reviewed. Pooled summary estimates were not calculated due to the variable methodology and PIS definitions. Tests performed on vaginal specimens accomplished similar performance to cervical and urine specimens for chlamydia (range of performance estimates: vaginal 65%–100%, cervical 59%–97%, urine 57%–100%) and gonorrhoea (vaginal 64%–100%, cervical 85%–100%, urine 67%–94%). Vaginal specimens were estimated to have a performance >80% for chlamydia and gonorrhoea infections in all but one study.

**Conclusions** Performance of the NAATs for chlamydia and gonorrhoea detection using vaginal specimens was similar to that of cervical and urine specimens relative to PIS. As vaginal samples have a higher acceptability and lower cost, the study can support clinical testing guidelines by providing evidence that vaginal samples are a suitable alternative to traditionally used specimens.

## Strengths and limitations of this study

► The systematic review included studies which had used a conservative definition of patient infection status (PIS), which can improve estimates of performance.

► The study included a broad time frame and the search did not have language restrictions, which increased the scope of the literature search.

► The lack of standardised definitions of PIS and variable methodologies used restricted us to qualitative analyses.

Nucleic acid amplification tests (NAATs) are recommended for the diagnosis of chlamydia and gonorrhoea due to their high sensitivity and specificity,[1–3] but a variety of urogenital specimens (ie, urine, cervical and vaginal) are used for testing at clinic level. The performance of NAATs to detect infection is known to vary by the type of specimen that is tested. A systematic review by Cook *et al*[4] estimated that the overall performance of NAATs for gonorrhoea was just 55.6% (95% CI 36.3% to 74.9%) for urine samples compared with 94.2% (90.5% to 98.0%) for cervical samples, with a specificity of 98.7% (97.5% to 99.9%) and 99.2% (98.4% to 100%), respectively.[4] Sensitivity is typically defined as the probability that a diagnostic test provides a positive result given that the the true test result—using gold standard—is positive, and a specificity is the probability that a diagnostic test yields a negative result given the true test result—using gold standard—is negative.

Recent studies have suggested that vaginal swabs may be better at detecting chlamydia and gonorrhoea than the traditionally used

## INTRODUCTION

In the UK, practices for *Chlamydia trachomatis* (chlamydia) and *Neisseria gonorrhoea* (gonorrhoea) testing are not currently standardised.

urine and cervical samples.[5–14] An issue related to these studies is that the estimates may be overestimated due to the use of discrepant analysis. As there is no gold standard diagnostic test for either chlamydia or gonorrhoea (ie, one with 100% sensitivity and specificity), studies evaluating the performance of a novel NAAT have tended to compare its performance against presumably poorer performing existing NAATs. Given the assumption that the novel NAAT has a higher sensitivity, it would be reasonable to expect that it would yield a positive result in some cases where the existing NAAT yields a negative result. The discrepant results create a challenge in differentiating false positive results from true positive results for the novel NAAT. In this situation, a third test is often performed. If this third test is positive, then the positive result from the novel NAAT is defined a 'true positive', and if the third test is negative, the result from the novel NAAT is defined a 'false positive'. This approach is called discrepant analysis (DA). DA has been criticised because it only retests 'false positives' and does not seek to improve the classification of 'false negatives' which potentially introduces a bias towards overestimating the performance of the novel NAAT.[15]

To address some of the potential bias from DA, patient infection status (PIS) was developed to represent an 'extended gold-standard' by measuring test performance based on information from multiple sources. The performance of PIS can be thought of as the sensitivity of a NAAT from a given site given the patient is considered to have an infection (determined by the sites and definition used for PIS). There is no consensus definition for PIS, but in general, it involves performing ≥1 test(s) at >1 site and interpreting the results using an algorithm that describes the number of positive tests (by test type or site) needed to define a person as infected. A limitation of the current evidence base is that many studies that have concluded in favour of using vaginal samples over cervical or urine for the diagnosis of chlamydia and gonorrhoea have used just one NAAT per test site.[6 9–14] Ideally, PIS is determined with the results of multiple NAATs adjusting for potential problems with the NAATs as well as for the possibility that a person might be infected at one or more anatomical sites.[16] PIS is not a perfect reference standard, and it has received criticism for similar reasons as discrepant analysis whereby there is a conditional dependence in the estimation of test performance and the PIS.[17] This may result in overestimation of the sensitivity of a test from a given site, but by including multiple sites in the PIS, it can also result in lower site-specific performance. PIS allows us to compare different sites within a study and similar PIS definition allows for comparison between studies. The ultimate use depends on the best combination of specimen and test. In most countries, health funding restraints mean it is only possible to use one specimen for chlamydia and gonorrhoea testing in women, and therefore, it is important to know which sites to sample to diagnose the highest proportion of infections.

The overall comparison of vaginal test performance compared with cervical and urine tests warrants further evaluation to inform evidence-based clinical testing guidelines. To this end, we performed a systematic review of the published literature. The primary aim of this review was to compare the performance of specimens taken from the vagina to those from the cervix or urine in diagnosing chlamydia and gonorrhoea infection in women based on a PIS definition. Given the lack of calibrated gold standard diagnostic tests and variation in the definition of PIS in the literature, we purposively used conservative selection criteria to identify studies that applied rigorous testing criteria in the process of defining PIS. We also examined the quality of the studies in light of the known biases in the PIS methodology to increase our ability to assess the evidence base.

## METHODS

We performed a systematic review following the Preferred Reporting Items for Systematic Reviews and Meta-Analyses (PRISMA) guidelines.[18] The protocol for our review was not registered in advance. We included studies which (1) tested for urogenital chlamydia and/or gonorrhoea infection in (2) postpubertal women (3) using a vaginal specimen (collected by any method) and (4) at least one other urine or cervical specimen. We limited this review to studies that used a conservative approach for defining PIS whereby each specimen site had to be tested with (5) ≥2 NAATs for chlamydia and ≥2 NAATs or 1 NAAT plus culture for gonorrhoea (as culture is an accepted alternative to NAAT for gonorrhoea diagnosis[19]). Finally, studies had to (6) report the performance of vaginal specimens compared with the PIS, or give the numbers of patients infected and results from the tests so that this could be calculated.

NAATs were defined as commercially available or in-house tests involving DNA or RNA replication using PCR, transcription-mediated amplification (TMA), ligase chain reaction (LCR) or strand displacement amplification (SDA). Studies that used pooled samples or novel point-of-care tests were excluded together with studies that used one NAAT on all samples and only confirmed positive or equivocal results with a second NAAT. No restrictions were set on year or language of publication. However, only peer-reviewed studies were included; conference abstracts/proceedings, notes, letters and book chapters were excluded.

The electronic databases EMBASE (1947 to 2017) and Ovid MEDLINE (1946 to 2017) were searched on 3 October 2017 by two authors, using OvidSP as the search platform (MMR, LMG-L or BD). The search strategy included keywords related to the infections, interventions, specimens of interest and outcomes. The full search strategy is presented in the supplement (see online supplementary file S1).

After duplicate references had been removed by OvidSP, two authors (MMR LMG-L, or BD) independently

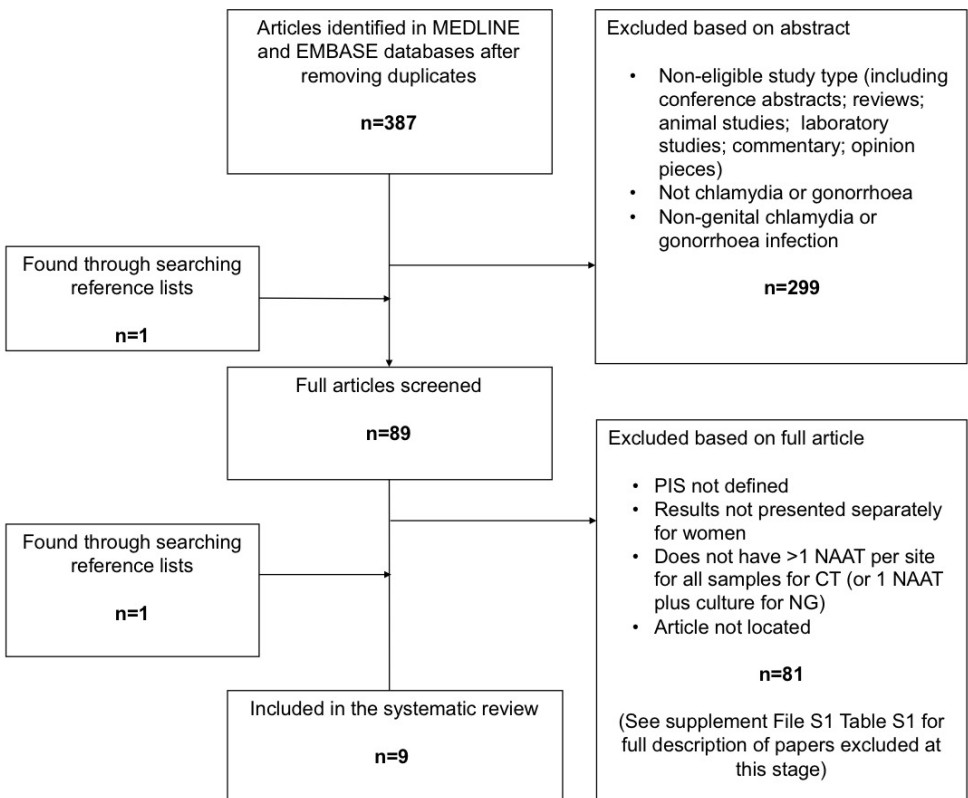

**Figure 1** Flow chart of study selection. CT, *Chlamydia trachomatis*; NAAT, nucleic acid amplification test; NG, *Neisseria gonorrhoea*; PIS, patient infection status.

selected studies by scanning titles and abstracts. A list of potentially eligible publications was obtained through consensus agreement and full texts were obtained online, through the British Library or directly from the authors. The full texts were then reviewed for inclusion and a third author was consulted if there was disagreement between the two reviewers. Where the language of the article was not English, we consulted colleagues fluent in the relevant language. The reference lists of all studies meeting the inclusion criteria, as well as recent key papers and systematic reviews identified during the search process, were examined iteratively for additional references.

One author extracted information from the selected studies using a data extraction form in Microsoft Excel. The extracted data were checked against the studies by a second author. Two reviewers independently assessed the risk of bias in patient selection, PIS definition and flow and timing (of different tests and analysis, including here also discrepant analysis) for each included study using an adapted version of the quality assessment of diagnostic accuracy studies (QUADAS-2) tool.[20] The purpose of this quality assessment was to consider the potential biases and variation within the individual studies that may account for heterogeneity across the studies. For each study, we calculated or reported the sensitivity of each NAAT for each sample site, and calculated CIs using the exact binomial distribution. If the study did not present raw numbers, the summary estimate was extracted instead. Performance was defined as the 'number of tests

appearing to be a true positive at a site for a given NAAT/ number of infected patients as determined by PIS'. These performance measures were used to compare the ability of each sample site to accurately predict PIS by describing the overall differences within and between sample sites for chlamydia and gonorrhoea. Analyses and forest plots were done in R.[21]

### Patient and public involvement

Vaginal specimens are an acceptable and sometimes a preferred way for chlamydia and gonorrhoea testing among women.[22] In this study, we compare the evidence base of performance of different testing sites for detection of chlamydia and gonorrhoea by conducting a systematic review of published primary research. We did not involve patient groups in the design or conduct of the study. The findings of the study can be used to standardise testing practices.

### RESULTS

The search on 3 October 2017 identified 567 studies of which 387 remained after deduplication by the search platform OvidSP. We excluded 299 studies based on their abstracts and screened the full text of the remaining 89 studies of which one was identified through searching reference lists. We excluded 81 of these studies and the reasons for exclusion are presented in the supplement (see online supplementary file S1, table S1). We identified

**Table 1**  Characteristics of included studies

| Author and year (Reference) // Included infections | Review outcome the main focus of the study? // Aim of the study | Tests performed, (acronym for the NAAT) | Specimens/sites (in women) // Comments | PIS definition // Comments on discrepant analysis |
|---|---|---|---|---|
| Chernesky M et al[27] (2006) CT | No To compare detection thresholds and inhibitor and infection rates from different specimens | APTIMA Combo 2, (TMA AC2) ProbeTec ET, (SDA ProbeTec ET) Amplicor, (PCR AMP) | 3 EC, 3 CCVS and FCU divided into three aliquots Each sample was tested as spiked (with added CT) or without added CT; seemingly all the tests were done to all the samples | ≥1 site positive by ≥2 different tests or two specimens positive in a single test (≥2 out of 9 specimen-tests done); VS included in PIS No discrepant analysis described |
| Cherneskey M et al[31] (2014) CT & NG | No To compare the performance of four second-generation NAATs with FCU and VS | APTIMA Combo 2, (TMA AC2) RealTime CT/NG, (PCR RealTime) ProbeTec CT/GC Q$^x$, (SDA ProbeTec CT/GC Q$^x$) Cobas CT/NG, (PCR Cobas CT/NG) | 1 FCU and 4 SCVS Each sample was tested as spiked (with added CT) or without added CT; all the tests were done to all the samples | ≥2 of the four assays were positive for any specimen type; VS included in PIS No discrepant analysis described |
| Cosentino LA et al[28] (2003) CT & NG | Yes To compare vaginal swab specimens to endocervical swab specimens for the detection of CT and NG | ProbeTec ET, (SDA ProbeTec ET) Amplicor, (PCR AMP) Thayer-Martin medium and chocolate agar (for NG culture) LCx, (LCR) for discrepant results | 3 EC, 2 (presumably clinician-obtained) VS. One EC used for NG culture CT was tested by PCR and SDA; NG (EC) was tested for by culture and SDA | Positive result by two different molecular tests for CT or for NG by culture or by two molecular tests; VS included in PIS Discrepant analysis for discordant results by LCR |
| Gaydos CA et al[25] (2010) CT & NG | No To compare the performance of the new RealTime CT/NG assay with the Aptima Combo two assay | RealTime CT/NG, (PCR RealTime) – index APTIMA Combo 2, (TMA AC2) – reference ProbeTec ET, (SDA ProbeTec ET) – reference NG culture | 4 EC, 1 SCVS, 2 CCVS, three urine Only one NAAT performed on the SCVS (this test was not used to define PIS); results for this are not presented | ≥1 positive result by both of the two reference NAATs, additionally for NG if culture positive the subject was defined as infected. Infection absent if ≥1 reference NAAT was negative for all sample types Discrepant analysis: for CT retested discordant results, for NG not done |
| Hook, E et al[23] (1997) NG | Yes To evaluate patient-obtained vaginal specimens tested with culture and LCR assays for NG compared with clinician-collected specimens | LCx, (LCR) Modified Thayer-Martin medium (for NG culture) | 3 SCVS, 3 EC (one sample at each site not part of this study as processed for CT) | Culture positive from either site; or LCR positive and culture negative with a positive confirmatory LCR; VS is included in PIS Discrepant analysis with, alternative TMA with different target site to confirm discordant results |
| Le Roy C et al[24] (2012) CT & NG* | No, data extracted based on their reporting Determine clinical performance of Bio-Rad CT/NG/MG assay for detection of CT, NG and *Mycoplasma genitalium* | Dx CT/NG/MG Assay, (qPCR) – index test Cobas TaqMan CT, (qPCR TaqMan) - reference NG culture | Symptomatic: 2 SCVS, 2 EC and 2 FCU. Asymptomatic: 2 SCVS and FCU. More tests done on symptomatic patients, but all samples seem to have been treated the same. | Study definition: At least two positive results from either of the two assays. We determined PIS based on test results for FCU and VS (which were available for all patients, see online supplementary material for further information); all infected patients had ≥2 positive tests and a positive test at both sites; VS is included in PIS Discrepant analysis used for discordant results |

**Table 1** Continued

| Author and year (Reference) // Included infections | Review outcome the main focus of the study? // Aim of the study | Tests performed, (acronym for the NAAT) | Specimens/sites (in women) // Comments | PIS definition // Comments on discrepant analysis |
|---|---|---|---|---|
| Schachter J et al[30] (2005) CT & NG | Yes To evaluate the performance of APTIMA assays on vaginal swabs for CT and NG | APTIMA CT, (TMA ACT) APTIMA GC, (TMA AGC) APTIMA Combo 2, (TMA AC2) ProbeTec ET, (SDA ProbeTec ET) | 1 FCU, 1 SCVS, 1 CCVS, 2 EC swabs All samples tested with three TMAs (two for CT and two for NG) FCU and 1x EC swab were also tested with ProbeTec ET) | Infected if either BD or AC2 were positive on FCU or EC. VS not included in PIS No discrepant analysis described |
| Shipitsyna E et al[26] (2008) CT | No To evaluate the performance of five PCRs and a recently introduced nucleic acid sequence-based amplification (NASBA) assay | Different 'in-house' PCRs tested: cPCR-DT, rtPCR-DT, cPCR-Ly, cPCR-Ep, rtPCR-Ep, real-time NASBA assay Reference methods: Amplicor, (PCR AMP) LightMix, (PCR Lightmix) | Used the subsample who had four specimens collected each: 2 EC and 2 VS All sample sites tested using reference NAAT and at least three other NAATs | Several Russian PCRs used on the sample and the sample was considered true positive if a positive result by a Russian PCR was confirmed by the reference tests; VS included in PIS Discrepant analysis for discordant results |
| Stary A et al[29] (1998) CT | No To compare TMA assay with LCR assay in detection of CT in genital tract with different specimen types | Aptima CT, (TMA ACT) LCx, (LCR) McCoy cell culture Direct-fluorescent antibody assay (DFA) or alternative TMA for confirming discrepant results | 3 EC, 3 CCVS, FCU EC and VS tested with LCR, TMA and culture. FCU tested with LCR and TMA | Positive culture at any site or positive result by both NAATs in one site, or one positive NAAT confirmed with discrepant analysis; VS included in PIS. Discrepant analysis with DFA or TMA with different target site to confirm discordant results |

Discrepant analysis : We define discrepant analysis to have occurred in situations where sample was positive for only one of the tests used. In these instances another test was done.
*Too few infections with gonorrhoea for analysis.
Tests used:LCR; PCR AMP (PCR Amplicor); PCR Cobas CT/NG (PCR Cobas CT/NG); PCR 'In house'; PCR RealTime (PCR RealTime); qPCR TaqMan (Quantitative PCR TaqMan); qPCR DxCT/NG/MG (Quantitative PCR DxCT/NG/MG); PCR LightMix, (PCR Lightmix); NASBA; SDA ProbeTec ET; SDA ProbeTec CT/GC Qx; TMA AC2; TMA ACT; TMA AGC; CT; NG; CCVS; EC; FCU; SCVS; VS.
AMP, Amplicor; CCVS, clinician-collected vaginal specimens; CT, *Chlamydia trachomatis*; EC, endocervical swabs; FCU, first catch urine; LCR, ligase chain reaction; NASBA, nucleic acid sequence-based amplification; NG, *Neisseria gonorrhoea*; PIS, patient infection status; qPCR, quantitative PCR; SCVS, self-collected vaginal specimens; SDA, strand displacement amplification; TMA AC2, transcription-mediated amplification Aptima Combo-2; TMA ACT, transcription-mediated amplification Aptima CT; TMA AGC, transcription-mediated amplification Aptima GC; VS, vaginal specimen.

one additional study through searching the reference lists[23] so that in total, nine eligible studies were included in this review: eight for chlamydia[24–31] and six for gonorrhoea.[23 25 28 30 31] A flowchart describing study selection is presented in figure 1 and a description of the included studies is presented in table 1. Few studies included results for patient characteristics or sample collection: three studies included self-collected vaginal swabs[23 30 31] and just one reported results for asymptomatic and symptomatic patients separately.[25]

The factors considered in the modified QUADAS-2 assessment and the results are presented in the supplement (see online supplementary file S1, table S2). All studies were rated as high or unclear risk for at least two of the five criteria related to risk of bias and applicability concerns; for example, risk for patient selection bias was either unclear (6/9) or high (3/9). Our inclusion criteria required there to be ≥2 NAATs for chlamydia and ≥2 NAATs or 1 NAAT plus culture for

gonorrhoea performed at each anatomical site, but variation remained in the number and type of tests used in the PIS algorithms. The definitions included '≥2 positive tests out of a number of tests performed (regardless of the site)', '≥2 different tests positive at the same site', 'any test positive in ≥2 different sites' and '≥2 different tests positive' (described in table 1). In all but one,[30] the risk of bias in the definition of PIS was considered high because the algorithm used was unclearly defined and/or included vaginal specimens (as the component whose performance is being evaluated should be independent of the gold standard it is measured against).

Seven out of nine studies (with the exception of Chernesky et al[27] and Chernesky et al[31]) had performed a different number of tests on a subsample of their study population where most of the studies (6/9) performed discrepant analysis for discordant results (eg, for sites where only one NAAT of many was positive). Due to the low number of studies identified, and their variable

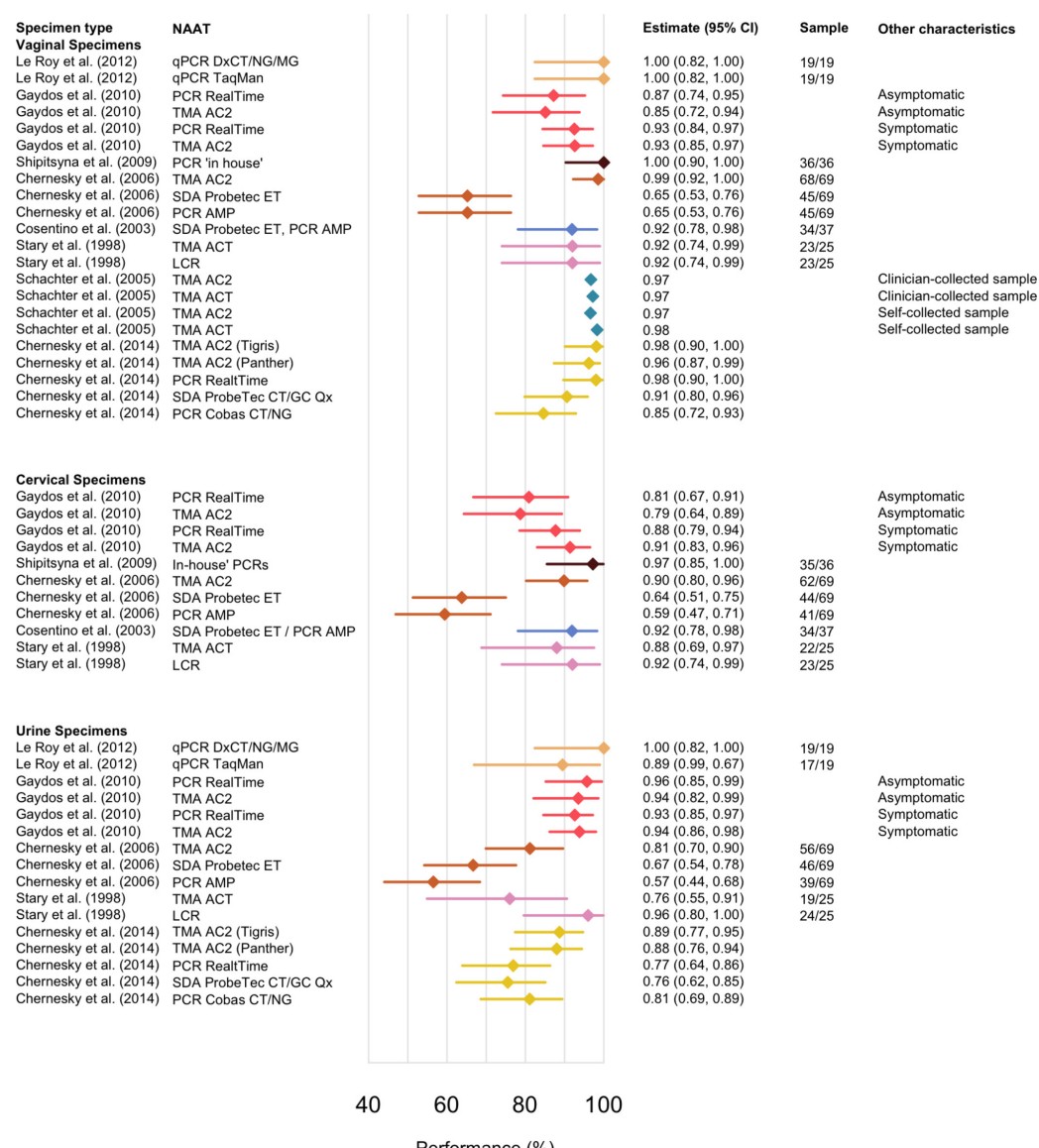

**Figure 2** Performance estimates for chlamydia by showing the sensitivity of the nucleic acid amplification test (NAAT) on a given specimen site relative to PIS. The studies are referenced by the first author, and publication year, and colours are by study for clarity. Each study is represented by a colour and where results are stratified by test, sampling or symptom status, this has been included in the figure. Tests used: LCR (ligase chain reaction); PCR AMP (PCR Ampicor); PCR Cobas CT/NG (PCR Cobas CT/NG); PCR 'In house'; PCR RealTi*me* (PCR RealTi*me*); qPCR TaqMan (quantitative PCR TaqMan); qPCR DxCT/NG/MG (quantitative PCR DxCT/NG/MG); PCR LightMix, (PCR Lightmix); NASBA (nucleic acid sequence-based amplification); SDA ProbeTec ET (strand displacement amplification Probetec ET); SDA ProbeTec CT/GC Q$^x$ (strand displacement amplification ProbeTec CT/GC Q$^x$); TMA AC2 (transcription-mediated amplification Aptima Combo-2); TMA ACT (transcription-mediated amplification Aptima CT); TMA AGC (transcription-mediated amplification Aptima GC).

methods and definitions of PIS, we deemed pooling of the results to be inappropriate. Instead, we compared the ability of each sample site to predict PIS using forest plots of site-specific performance from each study by NAAT used.

### Chlamydia

Of the eight studies that considered chlamydia test performance,[24–31] two directly addressed the question in this review[28 30] and the remainder aimed to test the sensitivity of a novel NAAT in a clinical setting.[24–27 29 31] All eight studies were performed in high-income countries. Where

a broader description of the population sample was reported, the studies had recruited patients from sexual health or reproductive health clinics, youth centres or primary care settings. One study specified that the sample population was high risk for acquisition of sexually transmitted infections.[29]

The reported chlamydia prevalence in the included studies ranged from 8.1% to 23.2% based on varying definitions of PIS. The number of infected women according to PIS ranged from 19 to 69 (with one study giving a point estimate without the raw data).[30] The site-specific

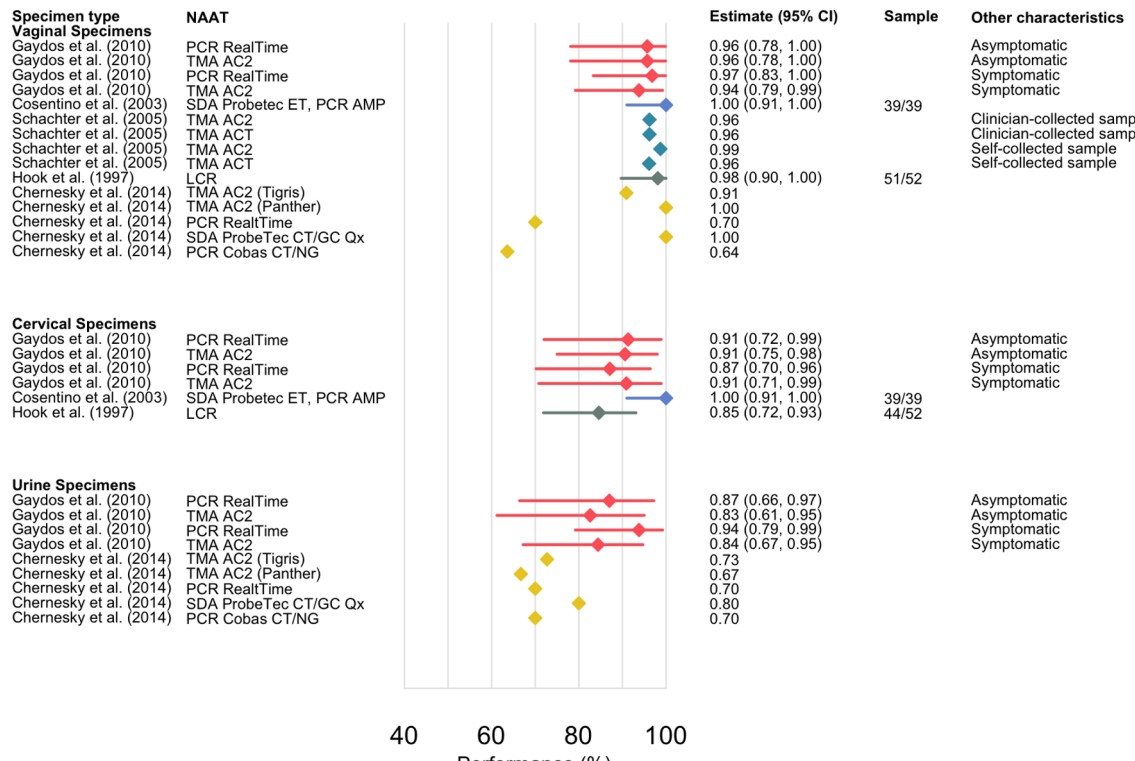

**Figure 3** Performance estimates for gonorrhoea showing the sensitivity of the nucleic acid amplification test (NAAT) on a given specimen site relative to patient infection status (PIS). The studies are referenced by the first author, and publication year, and colours are by study for clarity. Each study is represented by a colour and where results are stratified by test, sampling or symptom status, this has been included in the figure. Tests used: LCR (ligase chain reaction); PCR AMP (PCR Ampicor); PCR Cobas CT/NG (PCR Cobas CT/NG); PCR 'In house'; PCR RealTi*me* (PCR RealTi*me*); qPCR TaqMan (quantitative PCR TaqMan); qPCR DxCT/NG/MG (quantitative PCR DxCT/NG/MG); PCR LightMix, (PCR Lightmix); NASBA (nucleic acid sequence-based amplification); SDA ProbeTec ET (strand displacement amplification Probetec ET); SDA ProbeTec CT/GC Q$^x$, (strand displacement amplification ProbeTec CT/GC Q$^x$); TMA AC2 (transcription-mediated amplification Aptima Combo-2); TMA ACT (transcription-mediated amplification Aptima CT); TMA AGC (transcription-mediated amplification Aptima GC).

performance estimates and their CIs are presented in figure 2 and the results are presented in table format in the supplement (see online supplementary file S1, table S3). Across the studies, vaginal specimens performed similarly to cervical and urine specimens within a study, which can be observed by comparing results by study in figure 2 (performance range: vaginal 65%–100%, cervical 59%–97%, urine 57%–100%). The lowest overall performance estimates were reported by Chernesky *et al*[27] using a first-generation, less sensitive, NAAT.[27]

### Gonorrhoea

We identified six studies that tested for gonorrhoea.[23–25 28 30 31] Of these, five had a sufficient number of infected women according to PIS for performance to be calculated and were included in this analysis.[23 25 28 30 31] The excluded study found two gonorrhoea cases in 193 tested individuals and these were not analysed further.[24] Of the five remaining studies, three addressed the question in this review[23 28 30] while the remaining two compared the performance of a new assay.[25 31] As with chlamydia, all of the studies were performed in high-income countries.

The prevalence of gonorrhoea ranged from 2% to 16.8% in the five studies, based on PIS, as defined by

the studies; the denominator used for performance estimates varied from 11 to 52, with one study only reporting point estimates.[30] Figure 3 presents the forest plot and the results are presented in table format in the supplement (see online supplementary file S1, table S4). The performance estimates were similar within each study: performance range for vaginal samples was 63.6%–100%, for cervical samples 85%–100% and for urine samples 66.7%–94%. Three studies[23 25 28] included performance estimates for cervical swabs and two studies had performance estimates for urine.[25 31] The lowest overall performance estimates were reported by Chernesky *et al*.[31]

### DISCUSSION

We conducted a systematic review to evaluate the evidence for the relative performance of vaginal specimens in the detection of chlamydia and gonorrhoea in women using studies with conservative definitions of PIS. In the studies included in this review, we found similar sensitivities between urine, cervical and vaginal specimen types. This finding is contrary to other studies which have found that vaginal specimens have a higher performance than

cervical swabs.[5 7 9 32] This difference may be due to the less conservative definition of PIS used in other studies, which may be more prone for overestimation. Nevertheless, our study suggests that vaginal swabs are at least as good as urine and cervical swabs in detecting chlamydia and gonorrhoea.

This study followed PRISMA guidelines for systematic reviews and assessed the quality of the included studies using a modified version of QUADAS-2. The searches were conducted systematically and all potentially eligible abstracts were screened by at least two authors. The papers included in this review represent consensus opinion following group discussion among the authors. There were no language or time restrictions and multiple databases were searched. Therefore, this review represents a robust and comprehensive summary of extant evidence on the comparative performance of vaginal specimens in detecting chlamydia and gonorrhoea infection. A further strength of this study is that we excluded studies where only one test was performed at the vaginal site, even if ≥2 NAATs were used at all other sites.[6 10 33 34] By limiting the analysis to studies that performed ≥2 tests at multiple sites, we aimed to minimise the potential for overestimation of test performance.[16] The QUADAS-2 assessment found the studies were of variable quality and had methodological limitations and heterogeneity, particularly in PIS definition which was an important element of the analysis. As a result, some bias is likely to remain in the estimates reported in this review and the performance estimates should not be directly compared across studies. We did not pool the results, and considered the studies to be highly variable, which makes the study more exploratory in nature.

Overall, we found that vaginal specimens performed similarly to cervical and urine specimens for the diagnosis of both chlamydia and gonorrhoea in the identified studies. Based on the available evidence, and given the acceptability[22] and lower cost of vaginal samples,[8] our study further supports that vaginal samples provide an appropriate alternative to traditional cervical or urine testing methods. However, this review has also highlighted that there are a lack of studies that have attempted to explicitly answer this research question while taking into account the inherent biases in estimating diagnostic test performance, particularly for NAATs. Further primary research with appropriate robust methodology for determining test performance in symptomatic and asymptomatic patients is needed for a more authoritative comparison of the performance of different urogenital sample sites. We suggest that research should be directed towards applying more sophisticated statistical methods for comparing test performance such as latent class models,[17] which uses a weighted statistical model acknowledging the conditional dependence of the tests.

**Author affiliations**
[1]Department of Infectious Disease Epidemiology, Imperial College London, London, UK
[2]Department of Global Health and Population, Harvard T. H. Chan School of Public Health, Boston, Massachusetts, USA
[3]Leeds Sexual Health, Leeds Teaching Hospitals NHS Trust, Leeds, UK

**Acknowledgements** The authors would like to thank Dr Mikaela Smit and Dr Annick Borquez for their help in evaluating and translating articles not available in English.

**Contributors** HW and JDW conceived the study idea. MMR and LMG-L conducted the first review and BD and MMR conducted the update of the review with discrepancies solved by all three. MMR drafted the manuscript, which was subsequently edited by LMG-L, BD, HW and JDW. MMR prepared the tables and figures presented in the manuscript. All authors have read and accepted the final manuscript.

**Funding** MMR received funding from the Wellcome Trust (G090285/Z/09/Z) at the beginning of this study. BD has been funded by an MRC Population Health Scientist Fellowship (G0902120). HW, BD and LMG-L were supported in part by the National Institute for Health Research (NIHR) Biomedical Research Centre based at Imperial College Healthcare NHS Trust and Imperial College London. HW and JDW have received funding from the NIHR Research for Patient Benefit (RfPB) Programme.

**Disclaimer** The funders had no role in study design, data collection and analysis, decision to publish, or preparation of the manuscript.

**Competing interests** JDW has received research funding in the form of extra diagnostic reagents and equipment for one study from Hologic/Gen-Probe and has received one lecture fee from BD Diagnostics.

**Patient consent** Not required.

**Provenance and peer review** Not commissioned; externally peer reviewed.

**Data sharing statement** All data presented in this study is available in the manuscript or in the supplementary material.

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
