## [Reviewer comments · BMJ Open]

ARTICLE DETAILS

TITLE (PROVISIONAL)	Evaluation of the performance of nucleic acid amplification tests (NAATs) in detection of chlamydia and gonorrhoea infection in vaginal specimens relative to patient infection status: a systematic review
AUTHORS	Rönn, Minttu; Mc Grath-Lone, Louise; Davies, Bethan; Wilson, Janet; Ward, Helen

VERSION 1 – REVIEW

REVIEWER	Prof. Dennis V. Ferrero UNIVERSITY OF THE PACIFIC DEPARTMENT OF BIOLOGICAL SCIENCES 11881 NORTH ALPINE RD LODI, CA, USA
REVIEW RETURNED	20-Mar-2018

GENERAL COMMENTS	This paper represents a needed evaluation. For decades now, all evidence available tells us that self obtained vaginal swabs could be a useful tool in the public health arsenal; especially when one considers the lower financial cost of obtaining such samples and the ease of obtaining them (including non-medical settings). However, internationally, we have not advanced much beyond the traditional medical model of specimen collection for the most part. This paper should help move us along. The inclusion of the PRISMA checklist is useful information, but early on in the narrative, I suggest you define “PRISMA” by at least noting what the acronym stands for i.e. Preferred Reporting Items for Systematic reviews and Meta-Analyses. This is a robust analysis to tease out the important factors in identifying studies that are less biased. The authors’ study has resulted in an analysis that may finally move clinicians and public policy makers towards available and usable alternatives. Currently followed practices have for the most part stymied our movement to contain these two sexually acquired diseases. The authors’ discussion also helps to make us realize that it is not all about sensitivity and specificity; especially if we cannot even obtain a sample to test. The in-depth analysis of the two studies presented on pages 19-20 was most enlightening. This discussion highlights the problems inherent in single study analysis or “group think” and subsequent conclusion leading to public health policy challenges. Well done!
---

REVIEWER	Julius Schachter, prof Univ Calif, SF, CA, USA
REVIEW RETURNED	26-Mar-2018

GENERAL COMMENTS	This is a misguided survey of the literature done to generate an evidence based assessment of the sensitivity of vaginal swabs in
---

	diagnosis of chlamydial and gonococcal infection. This is misguided because vaginal swabs are specimens. Sensitivity is a measure of performance of diagnostic tests, not of specimens. Obviously ultimate use depends on the best combination of specimen and test, but this study is essentially presenting the vaginal swab as the variable, not the test. They note that some of the performance estimates of vaginal swabs were done in "studies that aimed to evaluate a novel NAAT" but they ignore the implications of that and just present the sensitivities generated. A little historical context may help here. The first generation of commercially available NAATs appeared in the early to mid 1990s, and the second generation, with improved sensitivity appeared about 10 years later. The first reports of use of vaginal swabs as specimens were by Stary and Hook in the late 1990s and some further evaluations were done with first generation NAATs. On line 316 of the manuscript authors state "The lowest overall sensitivity estimates were reported by Chernesky et al" and the numbers (65%) are used to establish the sensitivity range for vaginal swabs. What is not discussed there is that in Chernesky's study there was a third NAAT performed on vaginal swabs from the same women and it had a sensitivity of 98%. The best performing assay was a second generation test while the other two were first generation assays, now retired because of this type of inferiority. The poor sensitivity had nothing to do with the specimen, it was the assay that was inadequate. And the poor numbers were based on comparison with the better assay. These numbers do not belong in any discussion of vaginal swab performance.. some specific comments.:  1. The title presents what I think is a fatal flaw for this ms. Vaginal specimens don't have sensitivity, diagnostic tests do. 2. L40. Specimens do not detect microorganisms....diagnostic tests do! 3. L. 58 I beg to differ, >80% is inadequate sensitivity. It would have to be >90% to get FDA clearance. 4. L. 90 15yo studies are hardly recent 5. L. 95 This interpretation as to why vaginal specimens were being tested is wrong. Since 2010 CDC has stated that vaginal swabs were the specimen of choice for CT and GC detection by NAATs. Thus all new tests had to determine performance with vaginal specimens. 6. L. 124 This ignores the fact that having multiple sites included in the PIS results in lower site specific sensitivity. 7. L. 128 ref 14 incorrect.....in STD 8. L. 135 discussion of PIS, and your use of a conservative definition yielding better results, is not clear. Some examples might help/ 9. L. 142 these aims are not listed in abstract. 10. L.259 discrepant analysis is usually defined as a different test applied to a specimen that had a negative result when the paired specimen was positive. Retesting of equivocal specimens has a different purpose. It is not discrepant analysis 11. L.265 And there is a clear reason. Chernesky was comparing a second generation NAAT to 2 older formats. The latter were expected to be less sensitive.
--	---

REVIEWER	Lin Zhang University of Minnesota, United States
REVIEW RETURNED	17-May-2018

GENERAL COMMENTS	The authors conducted a literature review to compare the sensitivity of specimens taken from the vagina to those from the cervix or urine in diagnosing chlamydia and gonorrhoea infection based on a conservative definition of patient infection status (PIS) among women. Following are my main comments. Questions:  1. The paper considers a clinically interesting question. However, I do not think the authors deeply explore the studies. For example, for the primary aim, I do not seem direct analytical comparison of the sensitivity of vagina specimens to those from other specimens. In particular, many of the studies included in the review have sensitivities tested for both vagina and other specimens. It would be interesting to conduct a paired comparison between sensitivities of vagina specimen versus those of others across the multiple studies. 2. Page 6: Since there are no data to evaluate the difference in sensitivity of vaginal specimens from asymptomatic versus symptomatic patients and that were self- or clinician-collected, I don't think they should be included as secondary aims, but rather moved to discussion about future studies. 3. Pages 16 & 17: Compared to the range of sensitivity estimates, I think their distributions are of more interesting including the mean, median, IQR. 4. Page 19: Although PIS definitions are different, can the authors stratify the studies based on their PIS definition, and conduct sensitivity comparison within each stratum? 4. The captions of Figures 1&2 are not clear enough.
---

REVIEWER	Stephen Gerry University of Oxford, UK
REVIEW RETURNED	18-May-2018

GENERAL COMMENTS	I commend the authors for the efforts they have made in completing this review. I have a few comments: In the results section, I don't think there is much value in reporting the search results separately for the initial search and the updated search. Best to combine them. In the intro/methods sections a brief discussion of what sensitivity and specificity mean would be helpful. I'm not sure why specificity results have not been reported in the same fashion as sensitivity. From looking through a few of the papers I can see that the specificity is generally high, but I presume the number of negative results is much higher than the positive results, and therefore small changes in specificity may have a significant impact. Excuse my lack of knowledge, but its not clear to me whether the individual tests just report positive/negative or give a value within a range. If the latter, then the interpretation of sensitivities depends on the threshold chosen within each paper. Either way it might be helpful if this was clarified. Reporting the sample size and sensitivity (and CI) within the forest plots would be helpful.
---

VERSION 1 – AUTHOR RESPONSE

Reviewer(s)' Comments to Author:

Reviewer: 1

Reviewer Name: Prof. Dennis V. Ferrero

Institution and Country: UNIVERSITY OF THE PACIFIC
DEPARTMENT OF BIOLOGICAL SCIENCES
11881 NORTH ALPINE RD
LODI, CA USA

Please state any competing interests or state 'None declared': None Declared

Please leave your comments for the authors below

This paper represents a needed evaluation. For decades now, all evidence available tells us that self obtained vaginal swabs could be a useful tool in the public health arsenal; especially when one considers the lower financial cost of obtaining such samples and the ease of obtaining them (including non-medical settings). However, internationally, we have not advanced much beyond the traditional medical model of specimen collection for the most part. This paper should help move us along.

We thank the reviewer for their positive evaluation of the manuscript. We agree that the findings have relevance for the international public health community.

The inclusion of the PRISMA checklist is useful information, but early on in the narrative, I suggest you define "PRISMA" by at least noting what the acronym stands for i.e. Preferred Reporting Items for Systematic reviews and Meta-Analyses.

We have formatted this as suggested in the manuscript and to the supplemental material.

This is a robust analysis to tease out the important factors in identifying studies that are less biased. The authors' study has resulted in an analysis that may finally move clinicians and public policy makers towards available and usable alternatives. Currently followed practices have for the most part stymied our movement to contain these two sexually acquired diseases.

The authors' discussion also helps to make us realize that it is not all about sensitivity and specificity; especially if we cannot even obtain a sample to test. The in-depth analysis of the two studies presented on pages 19-20 was most enlightening. This discussion highlights the problems inherent in single study analysis or "group think" and subsequent conclusion leading to public health policy challenges. Well done!

We thank the reviewer for their positive feedback.

Reviewer: 2

Reviewer Name: Julius Schachter, prof

Institution and Country: Univ Calif, SF, CA
USA

Please state any competing interests or state 'None declared': current research funding from Abbott, BD and Roche,

Please leave your comments for the authors below

This is a misguided survey of the literature done to generate an evidence based assessment of the sensitivity of vaginal swabs in diagnosis of chlamydial and gonococcal infection. This is misguided because vaginal swabs are specimens. Sensitivity is a measure of performance of diagnostic tests, not of specimens. Obviously ultimate use depends on the best combination of specimen and test, but this study is essentially presenting the vaginal swab as the variable, not the test. They note that some of the performance estimates of vaginal swabs were done in "studies that aimed to evaluate a novel NAAT" but they ignore the implications of that and just present the sensitivities generated. A little historical context may help here. The first generation of commercially available NAATs appeared in the early to mid 1990s, and the second generation, with improved sensitivity appeared about 10 years later. The first reports of use of vaginal swabs as specimens were by Stry and Hook in the late 1990s and some further evaluations were done with first generation NAATs. On line 316 of the manuscript authors state "The lowest overall sensitivity estimates were reported by Chernesky et al" and the numbers (65%) are used to establish the sensitivity range for vaginal swabs. What is not discussed there is that in Chernesky's study there was a third NAAT performed on vaginal swabs from the same women and it had a sensitivity of 98%. The best performing assay was a second generation test while the other two were first generation assays, now retired because of this type of inferiority. The poor sensitivity had nothing to do with the specimen, it was the assay that was inadequate. And the poor numbers were based on comparison with the better assay. These numbers do not belong in any discussion of vaginal swab performance..

We thank the reviewer for taking the time to thoroughly review the manuscript. We agree with the reviewer's assessment on definitions, and our imprecise use of sensitivity, when considering the performance of vaginal specimens. In order to address this, we have corrected this throughout the manuscript.

In our reporting we have stratified all results by the test used, which as the reviewer highlighted is a measure of utility: "*ultimate use depends on the best combination of specimen and test*". We believe it is appropriate to include the results of first generation NAATs as although it is recognised that their sensitivity is lower than newer NAATs, any difference in performance using VVSs, compared with endocervical swabs and/or FCU, will still be evident as this will be dependant on the ability of the specimen to collect infected secretions. We have retained the estimates in our forest plots, but have removed a paragraph from our discussion.

We disagree the premise of the review is misguided, but we agree that our definitions can be improved.

some specific comments.:

1. The title presents what I think is a fatal flaw for this ms. Vaginal specimens don't have sensitivity, diagnostic tests do.

We have modified the title to: Evaluation of the performance of nucleic acid amplification tests (NAATs) in detection of chlamydia and gonorrhoea infection in vaginal specimens relative to patient infection status: a systematic review

2. L40. Specimens do not detect microorganisms....diagnostic tests do!

The language of the manuscript was changed to be more precise as described above. We revised this section as " We aimed to assess the performance of NAATs using vaginal specimens in comparison to other urogenital specimens in their ability to detect chlamydia and gonorrhoea infection in women."

3. L. 58 I beg to differ, >80% is inadequate sensitivity. It would have to be >90% to get FDA clearance.

We removed "high" from the abstract for a more conservative assessment.

4. L. 90 15yo studies are hardly recent

The line states "Recent studies have suggested that vaginal swabs may have a higher sensitivity for the detection of chlamydia and gonorrhoea than the traditionally-used urine and cervical samples (5–9)." The studies referenced are from 2003-2012 with only one study from 2003 and others published on 2008 or later. These were cited to represent the range of studies in the field with both UK and USA references included.

5. L. 95 This interpretation as to why vaginal specimens were being tested is wrong. Since 2010 CDC has stated that vaginal swabs were the specimen of choice for CT and GC detection by NAATs. Thus all new tests had to determine performance with vaginal specimens.

We have removed the sentence on line 95, and merged its references with the previous sentence.

6. L. 124 This ignores the fact that having multiple sites included in the PIS results in lower site specific sensitivity.

We have modified this as follows "This may result in overestimation of the sensitivity of a test from a given site, but it can also result in lower performance of a site as multiple sites are included in the PIS

definition. PIS allows us to compare different sites within a study and similar PIS definition allow for comparison between studies.”

7. L. 128 ref 14 incorrect.....in STD

We corrected the reference.

8. L. 135 discussion of PIS, and your use of a conservative definition yielding better results, is not clear. Some examples might help/

As the reviewer has commented, “ultimate use depends on the best combination of specimen and test” and that “by including multiple sites in the PIS results in lower site specific sensitivity.” In most countries, health funding restraints mean it is only possible to use one specimen for CT/NG testing in women so it is important to know which are the best sites to sample to diagnose the highest proportion of infections.

9. L. 142 these aims are not listed in abstract.

We removed the aims following reviewer 3, point 2.

10. L.259 discrepant analysis is usually defined as a different test applied to a specimen that had a negative result when the paired specimen was positive. Retesting of equivocal specimens has a different purpose. It is not discrepant analysis

We have made the language more precise and modified the section as follows “performed discrepant analysis for discordant results “ and below table 1 have added the following clarification:

Discrepant analysis: We define discrepant analysis to have occurred in situations where sample was positive for only one of the tests used. In these instances another test was done for the samples that were potentially positive.

11. L.265 And there is a clear reason. Chernesky was comparing a second generation NAAT to 2 older formats. The latter were expected to be less sensitive.

We have modified the text as follows: “The lowest overall performance estimates were reported by Chernesky *et al* using a first generation, less sensitive, NAAT”

Reviewer: 3

Reviewer Name: Lin Zhang

Institution and Country: University of Minnesota, United States

Please state any competing interests or state 'None declared': None declared

Please leave your comments for the authors below

The authors conducted a literature review to compare the sensitivity of specimens taken from the vagina to those from the cervix or urine in diagnosing chlamydia and gonorrhea infection based on a conservative definition of patient infection status (PIS) among women. Following are my main comments.

Questions:

1. The paper considers a clinically interesting question. However, I do not think the authors deeply explore the studies. For example, for the primary aim, I do not seem direct analytical comparison of the sensitivity of vagina specimens to those from other specimens. In particular, many of the studies included in the review have sensitivities tested for both vagina and other specimens. It would be interesting to conduct a paired comparison between sensitivities of vagina specimen versus those of others across the multiple studies.

We thank the reviewer for taking the time to review. We had hoped to carry out a more quantitative analysis at the start of the study, but given the systematic review resulted in fewer studies than expected with concerns about study quality in the included studies, we deemed quantitative comparison to not be appropriate.

2. Page 6: Since there are no data to evaluate the difference in sensitivity of vaginal specimens from asymptomatic versus symptomatic patients and that were self- or clinician-collected, I don't think they should be included as secondary aims, but rather moved to discussion about future studies.

We have removed the aims. And have added the following section to the discussion: "Further primary research with appropriate robust methodology for determining test performance in symptomatic and asymptomatic patients is needed for a more authoritative comparison of the performance of different urogenital sample sites. We suggest "

3. Pages 16 & 17: Compared to the range of sensitivity estimates, I think their distributions are of more interesting including the mean, median, IQR.

We have added more information to the forest plots, as suggested by reviewer 4. For each specimens and test result we have calculated a point estimate with a 95% confidence interval. To

obtain IQR or mean across the studies would require pooling of estimates, which we deemed inappropriate given the heterogeneity in the results.

4. Page 19: Although PIS definitions are different, can the authors stratify the studies based on their PIS definition, and conduct sensitivity comparison within each stratum?

The sample size is not sufficient to conduct stratification. There were no studies with identical PIS definitions.

4. The captions of Figures 1&2 are not clear enough.

We assume the reviewer is referring to figures 2 and 3 here (that describe the results) We have modified them, e.g. for figure 3: "Performance estimates for gonorrhoea showing the sensitivity of the NAAT on a given specimen site relative to PIS. The studies are referenced by the first author, and publication year, and colors are by study for clarity."

Reviewer: 4

Reviewer Name: Stephen Gerry

Institution and Country: University of Oxford, UK

Please state any competing interests or state 'None declared': None declared

Please leave your comments for the authors below

I commend the authors for the efforts they have made in completing this review.

I have a few comments:

In the results section, I don't think there is much value in reporting the search results separately for the initial search and the updated search. Best to combine them.

We have combined the reporting of the searches.

In the intro/methods sections a brief discussion of what sensitivity and specificity mean would be helpful.

We have added definitions to the introduction on the first paragraph: Sensitivity is typically defined as the probability that a diagnostic test provides a positive result given that the the true test result – using gold standard – is positive, and a specificity is the probability that a diagnostic test yields a negative result given the true test result – using gold standard – is negative.

I'm not sure why specificity results have not been reported in the same fashion as sensitivity. From looking through a few of the papers I can see that the specificity is generally high, but I presume the number of negative results is much higher than the positive results, and therefore small changes in specificity may have a significant impact.

The primary aim of the study was focused on the performance (defined as the sensitivity of the NAAT for a given site relative to PIS). Even though sensitivity may be overestimated, as outlined in the paper, there is some indication that the bias may be even larger for specificity, for a number of reasons but also due to the NAATs measuring presence of DNA instead of active infection (Hagdu Eur J Clin Microbiol Infect Dis (2009) 28:9–15)

Excuse my lack of knowledge, but its not clear to me whether the individual tests just report positive/negative or give a value within a range. If the latter, then the interpretation of sensitivities depends on the threshold chosen within each paper. Either way it might be helpful if this was clarified.

Each NAAT assay machine has inbuilt software with predetermined threshold values for each of the results. Hence the results are not open to individual interpretation and they cannot be influenced by the laboratory staff.

Reporting the sample size and sensitivity (and CI) within the forest plots would be helpful.

We have added the sample size, point estimate and 95% CI to the forest plots.

VERSION 2 – REVIEW

REVIEWER	Prof. Dennis V. Ferrero UNIVERSITY OF THE PACIFIC DEPARTMENT OF BIOLOGICAL SCIENCES 11881 NORTH ALPINE RD. LODI, CA USA
REVIEW RETURNED	21-Jul-2018

GENERAL COMMENTS	I thank the authors for their diligence as well as, their additions/changes in this manuscript. The current draft is better and will add to the knowledge base for CT/NG diagnostics and hopefully motivate public health practice in a better direction. As I said before, “Currently followed practices have for the most part stymied our movement to contain these two sexually acquired diseases”.
---

REVIEWER	Lin Zhang University of Minnesota, United States
REVIEW RETURNED	09-Aug-2018

GENERAL COMMENTS	The comparison of sensitivity of the NAAT on vaginal versus other specimens is more exploratory, without any formal statistical analysis. I think the authors should make this clear in conclusions and discussion.
---

VERSION 2 – AUTHOR RESPONSE

Reviewer(s)' Comments to Author:

Reviewer: 1

Reviewer Name: Prof. Dennis V. Ferrero

Institution and Country: UNIVERSITY OF THE PACIFIC, DEPARTMENT OF BIOLOGICAL SCIENCES, 11881 NORTH ALPINE RD. LODI, CA USA

Please state any competing interests or state 'None declared': None

Please leave your comments for the authors below

I thank the authors for their diligence as well as, their additions/changes in this manuscript. The current draft is better and will add to the knowledge base for CT/NG diagnostics and hopefully motivate public health practice in a better direction. As I said before, "Currently followed practices have for the most part stymied our movement to contain these two sexually acquired diseases".

We thank the reviewer for their positive comments.

Reviewer: 3

Reviewer Name: Lin Zhang

Institution and Country: University of Minnesota, United States

Please state any competing interests or state 'None declared': None declared

Please leave your comments for the authors below

The comparison of sensitivity of the NAAT on vaginal versus other specimens is more exploratory, without any formal statistical analysis. I think the authors should make this clear in conclusions and discussion.

We have added the following statements:

- Article Summary (page 3): The lack of standardised definitions of PIS and variable methodologies used restricted us to qualitative analyses.
- Article Discussion (page 19): We did not pool the results, and considered the studies to be highly variable, which makes the study more exploratory in nature.